



# Measurement report: New insights into the mixing structures of black carbon on the eastern Tibetan Plateau: soot redistribution and fractal dimension enhancement by liquid–liquid phase separation

Qi Yuan[1], Yuanyuan Wang[2], Yixin Chen[2], Siyao Yue[3, a], Jian Zhang[4], Yinxiao Zhang[5], Liang Xu[6], Wei Hu[7], Dantong Liu[2], Pingqing Fu[7], Huiwang Gao[1], Weijun Li[2*]

[1]Key Laboratory of Marine Environmental Science and Ecology, Ministry of Education of China, Ocean University of China, Qingdao 266100, Shandong, China
[2]Department of Atmospheric Sciences, School of Earth Sciences, Zhejiang University, Hangzhou 310027, Zhejiang, China
[3]State Key Laboratory of Atmospheric Boundary Layer Physics and Atmospheric Chemistry, Institute of Atmospheric Physics, Chinese Academy of Sciences, Beijing 100029, China
[4]School of Environmental and Material Engineering, Yantai University, Yantai 264005, Shandong, China
[5]Flight College, Binzhou University, Binzhou 256600, Shandong, China
[6]College of Sciences, China Jiliang University, Hangzhou 310018, Zhejiang, China
[7]Institute of Surface-Earth System Science, School of Earth System Science, Tianjin University, Tianjin 300072, China
[a]Now at: Max Planck Inst Chem, Minerva Res Grp, D-55128 Mainz, Germany

Correspondence: Weijun Li (liweijun@zju.edu.cn)

**Abstract.** Black carbon (BC, i.e., soot) absorbs radiation and contributes to glacier retreat over the Tibetan Plateau (TP). A lack of comprehensive understanding of the actual mixing state leads to large controversies in the climatic simulation of BC over the TP. In this study, ground-based sampling, electron microscopy analyses, and theoretical calculations were used to investigate the interactions among the liquid–liquid phase separation (LLPS), soot redistribution in secondary particles, and fractal dimension ($D_f$) of soot particles on the eastern rim of the TP. We found that more than half of the total analysed particles were soot-containing particles. One-third of soot-containing particles showed the LLPS phenomenon between organic matter and inorganic aerosols in individual particles, which further induced soot redistribution. The results show that a larger LLPS particle size, thicker organic coating, and smaller soot particles tended to drag soot from the sulfate core into the organic coating. The $D_f$ sequence is ranked as externally mixed soot (1.79±0.09) < sulfate-coated soot (1.84±0.07) < organic-coated soot (1.95±0.06). We concluded that the soot redistribution process and high RH both promoted the morphological compaction of soot particles. This study indicates that soot-containing particles experienced consistent ageing processes that induced a more compact morphology and soot redistribution in the LLPS particles on the remote eastern rim of the TP. Understanding the microscopic changes in aged soot particles could further improve the current climate models and evaluations of BC's radiative impacts on the eastern TP and similar remote air.



## 1 Introduction

The Tibetan Plateau (TP), often known as the "Third Pole", is the highest and largest plateau on Earth (Yao et al., 2012a). Due to the largest reserves of glacial area (~$10^5$ km$^2$) beside the polar regions, the TP acts as the "Asian water tower" and

plays a crucial role in supplying water for Asian ecosystems and human survival (Immerzeel et al., 2010). The eastern TP, as a source region of several large Asian rivers, has numerous glaciers but exhibits a serious glacial retreat (Yao et al., 2012b). Black carbon (BC, i.e., soot), the most important light-absorbing aerosol from biomass or fuel burning (Bond et al., 2013), is one of the major contributors to the retreat of eastern Tibetan glaciers through deposition on their surfaces (Xu et al., 2009;Li et al., 2016a;Hu et al., 2018;Shao et al., 2017;Dong et al., 2016;Li et al., 2018). Long-term field measurements indicate that

BC concentrations exhibit a high level in the eastern TP atmosphere and snow cover (Kang et al., 2022). A GEOS-Chem model with a 20-yr simulation has shown that approximately 35% of surface BC on the TP comes from East Asia, and a strong East Asian summer monsoon leads to large amounts of BC transported from central China (Han et al., 2020). BC from the Sichuan Basin can be transported to the eastern TP by an intensified southwesterly wind or can penetrate to the eastern TP in the planetary boundary layer (Yuan et al., 2020b). Therefore, the long-range transported BC from East Asia to

the eastern TP has become a hot spot in aerosol research (Kang et al., 2022).

In recent decades, many studies have focused on the climatic effects of BC over the TP (Chen et al., 2022b;Zhu et al., 2021). It is worth noting that the estimation of direct radiative forcing of BC exhibited a large difference in several model studies over the entire TP, such as a four-stream radiative transfer model simulation of +0.29–+2.61 W m$^{-2}$ (Kopacz et al., 2011), a global chemical transport model in conjunction with a radiative transfer model simulation of +2.3 W m$^{-2}$ with an uncertainty

of ~70–85% (He et al., 2014), a Community Atmosphere Model Version 5 simulation of +0.31±0.12 W m$^{-2}$ (Zhang et al., 2015), and an Santa Barbara DISORT Atmospheric Radiative Transfer model simulation of +4.6±2.4 W m$^{-2}$ (Liu et al., 2021). This uncertainty in BC radiative forcing is largely caused by the lensing effects of the coating, which are sensitive to the mixing structure (Li et al., 2016b). One of the key limitations is the variability of the actual BC-mixing state in most models and in the ambient air (Hu et al., 2021;Zhai et al., 2022;Fierce et al., 2020;Riemer et al., 2019;Adachi and Buseck,

2013), especially in different areas of the vast TP. For instance, Wang et al. (2017a) used measurement data to estimate that coatings (sum of organics, sulfate, nitrate, chloride, and ammonium) led to 40% light absorption enhancement of BC on the central TP. In contrast, Wang et al. (2018) applied the Mie model to simulate an average BC absorption enhancement of up to 1.9 on the southeastern TP. These large variations in BC absorption enhancement can be attributed to the apparent discrepancy and uncertainty between the model simulations and observations regarding the mixing structures of BC particles

(Yuan et al., 2019;Tan et al., 2021;Wang et al., 2021a;Zhao et al., 2017;Fu et al., 2022). Therefore, it is crucial to perform extensive in situ measurements and gain more observational data on mixing states of BC particles to improve the current understanding and simulation of the climatic effect of BC on the eastern TP.

Recently, Zhang et al. (2022) observed, for the first time, a unique mixing structure with multiple BC particles redistributed in organic coatings instead of sulfate cores at a high mountain site in central China. BC redistribution is induced by





liquid–liquid phase separation (LLPS) and can reduce the absorption enhancement effect by 28–34% (Zhang et al., 2022). This interesting finding draws our attention to the mixing structure of BC in similar mountainous areas on the eastern rim of the TP. This raises some new questions: is BC redistribution the case or norm in the ambient air? do BC particles undergo similar redistribution during long-range transport from central China to the eastern TP? If so, BC redistribution should be incorporated into the current climate models to correct the absorption enhancement of BC on the eastern TP. To date, there is

no evidence for revealing the redistribution mechanisms of soot particles during aerosol phase separation in eastern TP air. In addition, Yuan et al., (2019) revealed that sulfate coating under liquid phase can affect the morphology and compactness of the inner BC. However, can the organic coating after BC redistribution affect the morphology of the inner BC? The data scarcity serves as an obstacle towards a more accurate simulation in atmospheric models of their optical absorption on the eastern or the whole TP.

In this study, individual particle collection, transmission electron microscopy (TEM), and atomic force microscopy (AFM) were comprehensively employed to investigate the mixing structures of soot particles at a mountain site on the eastern fringe of the TP. Theoretical calculations were applied to obtain an important morphological parameter (fractal dimension, $D_f$) of soot particles in different mixing states. The principal concerns of this study are as follows: (1) Can the redistribution of soot particles occur in the eastern TP atmosphere? (2) What are the key factors that affect the redistribution process of soot

particles? (3) How do the morphologies of soot particles change after redistribution? The answers to these questions will be helpful to better understand the ageing processes of soot particles under LLPS. The findings can also promote the future evaluation of climatic and environmental effects of anthropogenic aerosols on the eastern TP.

## 2 Methodology

### 2.1 Sampling site and meteorological analysis

The sampling site was on the top of Mt. Emei (103.33°E, 29.52°N, 3048 m a.s.l. – above sea level), which is located on the eastern rim of the TP (Fig. 1a). Mt. Emei lies in a transitional zone from the low-altitude, urbanized, and industrial Sichuan Basin on the northern and eastern side to the high-altitude, rural, and remote TP on the western side (Fig. 1). Due to the unique mountain–basin topography and intense anthropogenic activities, the Sichuan Basin is one of the most heavily polluted regions in China (Zhang et al., 2012;Shu et al., 2022;Chen et al., 2022a). We obtained PM$_{2.5}$ and BC column mass

density in June-July 2016 from the Giovanni online data system (https://giovanni.gsfc.nasa.gov/giovanni/) (Fig. 1b-c). A high-emission zone in the Sichuan Basin was detected during the sampling period (Fig. 1b-c). Several field and model studies showed that large amounts of anthropogenic air pollutants from the Sichuan Basin could be transported to the eastern TP (Jia et al., 2019;Zhang et al., 2015). Mt. Emei has proven to be a suitable site for investigating the long-range transport of anthropogenic aerosols (Zhao et al., 2020).

Twenty-four-hour backwards trajectories were calculated every 1 h at an ending height of 100 m above ground level by using a Hybrid Single Particle Lagrange Integrated Trajectory (HYSPLIT) (Fig. 1a). According to the HYSPLIT calculation,





there were two dominant backwards trajectory types during the individual particle-sampling period: eastern trajectories from the Sichuan Basin (red trajectories in Fig. 1a) and western trajectories from the TP (blue trajectories in Fig. 1a). Various active fire spots were detected over South and Southeast Asia by the Fire Information for Resource Management System

(FIRMS) provided by the MODIS satellite (https://firms.modaps.eosdis.nasa.gov) (Fig. 1a). Meteorological data such as relative humidity (RH), temperature (T), wind speed (WS), and wind direction (WD) were measured and recorded every 5 min by an automated instrument (Kestrel 5500, USA) (Fig. S1).

## 2.2 Individual particle collection

A DKL-2 sampler (Genstar Electronic Technology, China) was used to collect individual aerosol particles on copper TEM

grids covered by carbon film (carbon type-B, 300-mesh copper; Tianld Co., China). The individual particle samples were collected according to the detailed procedures in Yuan et al. (2019). A total of 15 aerosol samples were collected in the summer of 2016 (22 June–09 July). The daily sampling times (Beijing) were 09:00, 15:00, and 22:00, with a sampling duration of 45 min at each time. Dry plastic capsules (100 mesh; Tianld Co., China) were used to place the TEM grids to prevent contamination. Afterwards, the samples were stored in a desiccator at 20-25% RH until analysis.

## 110 2.3 Electron microscopic analysis

The TEM (JEOL JEM-2100, Japan) was operated at 200 kV to analyse 15 aerosol samples by analysing 4499 aerosol particles to obtain their mixing state and morphology. An energy-dispersive X-ray spectrometer (EDS, INCA X-MaxN 80T, Oxford Instruments, UK) was used to detect elements with compositions heavier than C (Z≥6). TEM observations directly identified soot structure in most individual particles. However, for some typical soot with thick sulfate coating, we normally

put the particles under the electron beam for a longer time to destroy the sulfate coating. Then we further clearly identified the structure of soot core left in the TEM image. RADIUS software (EMSIS GmbH, Germany) was used to measure the morphological parameters of individual particles. Further details of TEM analysis can be found in Yuan et al. (2021).

The three-dimensional morphology of the aerosol particles was also studied by using an atomic force microscope (AFM, Dimension Icon, Veeco Instruments, Inc., USA) operated in tapping mode under ambient conditions. The bearing area and

bearing volume of each analysed particle can be obtained from the AFM images by Nanoscope analysis software (Chi et al., 2015). The equivalent circle diameter (ECD, d) and the equivalent volume diameter (EVD, D) were calculated according to the bearing area and bearing volume. As shown in Fig. S2, the relation between d and D is D=0.4144×d. The sizes and coating thicknesses of individual particles are calculated based on the EVD.

## 2.4 Fractal dimension of soot particles in different mixing states

The morphology and compactness of soot particles can be characterized by $D_f$, which is calculated by the scaling law below (Koylu et al., 1995). Several previous studies have applied this method to investigate the morphological variations of soot particles in different regions (Wang et al., 2021b; Yuan et al., 2019):


$$N = k_g \left( \frac{2R_g}{d_p} \right)^{D_f} \qquad (1)$$

where $N$ is the total number of soot monomers; $R_g$ is the radius of gyration of individual soot particles; $d_p$ is the average diameter of soot monomers; $k_g$ is the fractal prefactor; and $D_f$ is the mass fractal dimension of individual soot particles.

In addition to $D_f$, we also used convexity ($CV$), roundness ($RN$), and aspect ratio ($AR$) to quantify the morphology of soot. The $CV$ is a measure of the topological properties of the particle's projection and is the ratio of the projected area of the particle to the area of the convex hull polygon. $RN$ is a measure of the ratio of the projected area of the particle to the area of a circle of a diameter equal to the longest dimension. $AR$ is the maximum ratio between the length and width of a bounding box. These morphological parameters can be calculated using the equations in Yuan et al. (2019).

## 3 Results and discussion

### 3.1 Mixing states of soot particles on Mt. Emei

We analysed thousands of particles with the diameter at 30 nm–4 μm, and approximately 54.2% of the total particles were either externally or internally mixed soot particles, which we define as soot-containing particles (Figs. 2 and S3). The proportion of soot-containing particles was lower than our previous field observations at a rural site on the southeastern TP (64%) (Yuan et al., 2019) but higher than that at a remote site in the central Himalaya (51%) (Yuan et al., 2020a). We also found that the proportion of soot-containing particles with eastern basin trajectories prevailing (65.4%) was much higher than that during the dominance of the western TP trajectories (50.2%) (Fig. S3). The results indicate that regional anthropogenic emissions, especially from incomplete fossil-fuel burning in the Sichuan Basin, frequently influenced the upper air layer during the sampling period. A similar result was derived from a bulk measurement by Zhao et al. (2020).

Based on the morphologies and mixing states of individual soot-containing particles, we classified them into three major types: externally mixed soot (named soot), internally mixed soot with only sulfate (named S-soot), and core–shell particles with soot in the sulfate core or soot in the organic matter (OM) coating (named S-soot-OM-coating) (Fig. 2). Soot particles exhibit a unique chain-like aggregate morphology and contain C with minor O (Fig. 2a-b). Here, more than half of soot-containing particles were identified as internally mixed S-soot particles (Figs. 2c and 3a). Sulfate commonly displays some surface bubbles under an electron beam (Fig. S4a) and partly or entirely coats soot particles (Fig. 2c). In this study, high-resolution TEM further identified secondary sulfate particles coated by an OM-coating on Mt. Emei (Figs. 2e, 2g, S4b). The sulfate core and OM-coating in secondary particles were identified as LLPS (You et al., 2012;Zhang et al., 2022). Li et al. (2021) used cryo-TEM measurements to confirm that the core–shell particles observed by regular TEM can reflect the actual mixing state of individual particles in ambient air. Indeed, laboratory studies and field observations have proven that LLPS can induce a typical core–shell structure in secondary particles (Li et al., 2021;Zhang et al., 2022;You et al., 2012;Kirpes et al., 2022;O'Brien et al., 2015).



A laboratory study and field observations have shown that LLPS can drive soot in core–shell particles from inside inorganic aerosols to outer organic aerosols, which is called the soot redistribution phenomenon (Brunamonti et al., 2015;Zhang et al., 2022). TEM images show that the soot occurred either in the sulfate core (Fig. 2e) or in the OM-coating (Fig. 2g). Zhang et

al. (2022) applied cryo-TEM to prove that the dry state of the phase-separated soot particles by TEM analysis can represent the actual distribution of soot particles with LLPS under different RHs in ambient air. Figure 2e and 2g show that the LLPS particles contained tiny soot particles forming the S-soot-OM-coating particles. In this study, the redistribution of soot particles was observed for the first time in eastern TP air.

### 3.2 Relative abundance and size distribution of S-soot-OM-coating particle

Figure 3 displays the relative abundances of soot, S-soot, and S-soot-OM-coating particles in all soot-containing particles. The number contributions of S-soot-OM-coating particles to the total soot-containing particles ranged from 5.9% to 78.2% with an average value of 34.8% (Fig. 3a). The TEM image shows that large numbers of S-soot-OM-coating particles occurred in the sample collected at 10:00 on 23 June (Fig. 3b). The results show that large amounts of soot-containing particles were involved in the LLPS process during the sampling period.

Figure 4a shows that soot-containing particles generally occurred in the fine mode (<1 μm). The size distribution of soot and S-soot particles displayed the peak at 134 nm and 188 nm, respectively, indicating that a secondary sulfate coating enlarged soot-containing particle sizes (Fig. 4a). Figure 4b shows that the relative abundance of S-soot-OM-coating particles increased with increasing particle size and reached nearly 70-80% at particles larger than 500 nm. Therefore, particle size is an important factor affecting LLPS (Li et al., 2021). Here, it is necessary to investigate whether particle size can affect the

redistribution of soot particles in S-soot-OM-coating particles.

### 3.3 Redistribution of soot particles in OM-coatings

Figure 5a shows that 73% of the S-soot-OM-coating particles were soot in OM-coating particles (named soot-Ocoating particles for short). 24% of the S-soot-OM-coating particles were fractional soot in the inorganic sulfate core and other soot in OM-coating (named soot-Icore-Ocoating particles for short) (Fig. 5a). The remaining 3% of the S-soot-OM-coating

particles were all soot in the inorganic sulfate core (named soot-Icore particles for short) (Fig. 5a). The soot-Ocoating particle at 73% in this study was much higher than the reported 59% in the background air in northern and central China (Zhang et al., 2022). Therefore, we can conclude that soot redistribution in secondary particles is a common phenomenon on Mt. Emei. Zhang et al. (2022) suggested that the soot redistribution is mainly influenced by the ratio between the OM-coating thickness and the particle size. Similar to the method reported by Zhang et al. (2022), we calculated the OM-coating

thicknesses and the entire particle sizes (Fig. 5). We found that the soot-Ocoating particles mostly had larger sizes (avg. 790 nm) and thicker coatings (avg. 50 nm) than those of soot-Icore-Ocoating particles (avg. 737 nm for particle size and 35 nm for coating thickness) and soot-Icore particles (avg. 571 nm for particle size and 19 nm for coating thickness) (Fig. 5b-e).



Figure 5b shows that the entire particle size and coating thickness exhibited good correlations, suggesting that larger particles along with thicker OM-coatings can drive soot particles into the organics from the sulfate core due to LLPS.

We used the ratio of OM-coating thickness divided by the size of soot in the S-soot-OM-coating particles (OM/soot) to further examine the redistribution mechanisms of soot particles. Figure 6a shows that the ratios of OM/soot in soot-Ocoating particles were obviously higher (avg. 0.32) than those in soot-Icore-Ocoating particles (avg. 0.24) and soot-Icore particles (avg. 0.12). With the increasing ratio of OM/soot, soot tended to distribute in the organic phase (Fig. 6b). When the ratio was larger than 0.2, more than 80% of the internally mixed soot was distributed in the OM-coating, suggesting that soot was

more likely to be distributed in the organic phase than in the inorganic phase along with the incorporation of the OM-coating (Fig. 6b). Nearly all soot particles would occur in the OM-coating when the ratio of OM/soot was larger than 0.6 (Fig. 6b). The TEM images clearly showed the transferred position of soot from the inner sulfate core to the outer organic coatings with increasing OM/soot (Fig. 6c-e).

Here, we noticed that two or more soot particles were distributed in the OM-coating (Fig. 6d). During the sampling period,

approximately 27% of the total analysed S-soot-OM-coated particles contained one soot particle, while 73% of them captured two or more soot particles in the OM-coating (Fig. 7a). Figure 7b shows a good positive correlation between soot particles in the OM-coating and particle sizes. The results suggest that the coarser particles following the thicker OM-coatings captured more soot particles in the OM-coating during the redistribution process (Fig. 7b-e).

All of these observations provided direct in situ evidence for soot redistribution in eastern TP air. The soot redistribution is

probably governed by the entire particle size, OM-coating thickness, and soot size. In other words, soot particles tend to occur in organics with increasing particle size, increasing OM-coating thickness and decreasing soot size.

### 3.4 $D_f$ changes of soot particles after redistribution

To examine the specific morphological changes of soot particles, we calculated the $D_f$ of externally mixed soot, sulfate-coated soot and organic-coated soot. The average $D_f$ of externally mixed soot on Mt. Emei was 1.79±0.09 (Table 1), which

was slightly higher than that on the southeastern TP (1.75±0.08) (Yuan et al., 2019), suggesting that the sources of soot particles in the eastern TP atmosphere were more complex (Pang et al., 2022). Figure 8 and Table 1 show that the $D_f$ sequence from low to high values is ranked as externally mixed soot (1.79) < sulfate-coated soot (1.84) < organic-coated soot (1.95). Higher $D_f$ values reflected a more compacted structure of the highly aged soot particles by internally mixing with sulfate and organics (Yuan et al., 2019;Wang et al., 2017b;China et al., 2015). In addition to $D_f$, the variation in

morphological parameters such as $CV$, $RN$ and $AR$ can also indicate the morphological changes of individual soot particles on Mt. Emei (Table 1). The sulfate-coated soot and organic-coated soot particles had a higher $CV$ (0.87 and 0.87, respectively), higher RN (0.41 and 0.42, respectively) and lower $AR$ (1.61 and 1.61, respectively) than those of externally mixed soot (avg. $CV$=0.81, avg. $RN$=0.38, and avg. $AR$=1.63). The conclusion derived from all these morphological parameters is consistent with the compacted structure of aged soot particles by coating with sulfate and organics indicated by

$D_f$. We found a significant increase in $D_f$ when soot was coated with organics compared with sulfate-coated soot. This result





suggested that the redistribution process during LLPS can further increase the compactness of soot. Xue et al. (2009) found that soot embedded within low-viscosity organics could increase its compactness after coating.

The Mt. Emei site usually had an extremely humid environment with >90% RH in more than 60% noncloud sampling periods (Fig. S1). The high RH has been proven to contribute to the increase in soot compactness through the phase change

of coating aerosols (Yuan et al., 2019). Figure 8b-d shows the $D_f$ values of externally mixed soot, sulfate-coated soot, and organic-coated soot under RH>90% and RH<80%. We noticed that $D_f$ of organic-coated soot increased from 1.90 to 1.97 with increasing RH from 63%-76% to >90% (Fig. 8b). Sulfate-coated soot had a greater increase in $D_f$ from 1.76 to 1.99 with increasing RH from 63%-76% to >90% (Fig. 8c). The results proved that high RH could facilitate the aged soot particles to become more compact regardless of inorganic- or organic-coated materials. In view of the larger increase in

sulfate-coated soot, we can infer that atmospheric ageing via sulfate coating favours the compaction of soot due to the hygroscopic property of the sulfate coating.

## 4 Implications and summary

BC, as the most important light-absorbing aerosol, is a nonnegligible contributor to atmospheric warming over the Third Pole (Chen et al., 2022b;Zhang et al., 2021a;Cong et al., 2015;Zhang et al., 2021b). Due to the high BC concentration on the

eastern TP (Yuan et al., 2020b), it is necessary to give more attention to the climate-related factors of BC in eastern TP air, especially the mixing state and ageing processes. In this study, ground-based sampling, electron microscopy analyses and theoretical calculations were used to investigate the atmospheric processes of soot particles on the eastern rim of the TP.

We analysed thousands of individual particles, and more than half of the total particles were soot-containing particles. Electron microscopy analysis indicates that one-third of the total soot-containing particles exhibited a sulfate core–organic

shell structure with soot distributed in either the inorganic core or the organic coating. LLPS has been proven to be a key atmospheric process that contributes to the formation of this core–shell particle (Li et al., 2021). Approximately three quarters of the core–shell soot-containing particles experienced a redistribution process that drove soot from the inorganic phase to the organic phase. In summary, this study provides the first direct in situ evidence of the redistribution of soot particles in eastern TP air through electron microscopy analyses.

Long-range transported air masses from the highly polluted Sichuan Basin or interior TP could carry large amounts of anthropogenic pollutants to eastern TP air. The ageing process during long-range transport would result in an increase in the organic-coating thickness and entire particle sizes (Xu et al., 2020). Our morphological analysis suggests that the entire particle size and ratio of organic coating thickness divided by the size of soot are two key factors affecting the redistribution of soot. Larger particles with thicker organic coatings can drive and capture more smaller soot particles from the inorganic

phase to the organic phase.

Theoretical calculations have shown that aged soot particles internally mixed with sulfate have higher $D_f$ values (avg.±sdev: 1.84±0.07) than those of externally mixed soot (1.79±0.09). The redistribution process may strengthen the compaction of



soot particles with higher $D_f$ values of soot in the organic coatings (1.95±0.06). Moreover, high RH can promote the compaction of aged soot coated with either sulfate or organics. The assessment of the optical properties and climatic effects

of soot particles is highly dependent on the shape and position of the soot within the entire host particle (Adachi et al., 2010;Wang et al., 2021b;Zhang et al., 2022). It is important to quantify the key morphological parameters to serve the climatic simulation of BC in eastern TP air.

In summary, the results of this study will help to improve the understanding of the mixing structure and ageing process of BC in the TP atmosphere (Fig. 9). The observation of the redistribution of BC after LLPS may be a common phenomenon

over the entire TP, at least in highly anthropogenically influenced regions with high RH. These findings will promote the future evaluation of the climatic effects of BC at the Third Pole.

**Data availability.** Data presented in this paper are available from https://doi.org/10.6084/m9.figshare.21988439.

**Supplement.** The supplement related to this article is available online.

**Author contributions.** WL and QY designed the research. QY performed the data analysis and wrote the manuscript, and WL revised it. SY assisted with the sample collection. YW, YC, JZ, YZ and LX carried out the TEM analysis of individual particles. WH, DL, PF, and HG contributed to the improvement of this paper. All the authors approved the final version of

this paper.

**Competing interest.** At least one of the (co-)authors is a member of the editorial board of *Atmospheric Chemistry and Physics*. The peer-review process was guided by an independent editor, and the authors also have no other competing interests to declare.


**Disclaimer. Publisher's note:** Copernicus Publications remains neutral with regard to jurisdictional claims in published maps and institutional affiliations.

**Financial support.** This work was funded by National Natural Science Foundation of China (91844301, 42075096),

Zhejiang Provincial Natural Science Foundation of China (LZ19D050001, LY21D050002), the Research Funding of School of Earth Sciences of Zhejiang University and the Research Funding of Key Laboratory of Atmospheric Chemistry, China Meteorological Administration (2022B09)



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

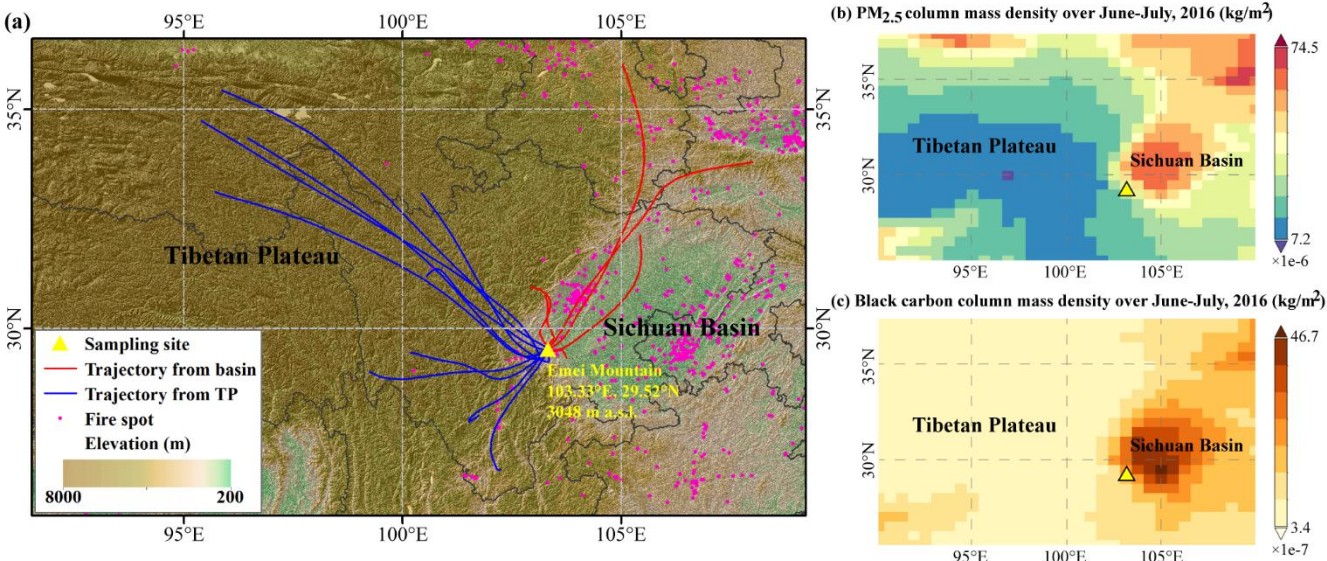


**Figure 1: Location of the sampling site, twenty-four-hour backwards trajectories during 22 June–9 July 2016, and PM$_{2.5}$ and BC column mass density covering the eastern TP and Sichuan Basin in June–July 2016. (a) Location of the sampling site with the Sichuan Basin on the eastern side and the TP on the western side (103.33°E, 29.52°N, 3048 m a.s.l. – above sea level). The twenty-four-hour backwards trajectories of each individual particle sample collection time were calculated using the Hybrid Single**
**Particle Lagrange Integrated Trajectory model (HYSPLIT). Blue lines represent the western TP trajectories, and red lines represent the eastern basin trajectories. (b-c) PM$_{2.5}$ and BC column mass density covering the eastern TP and Sichuan Basin in June–July 2016 (Giovanni online data system, https://giovanni.gsfc.nasa.gov/giovanni/).**

2023-02-01
Atmospheric Chemistry and Physics
10.5194/acp-2022-831



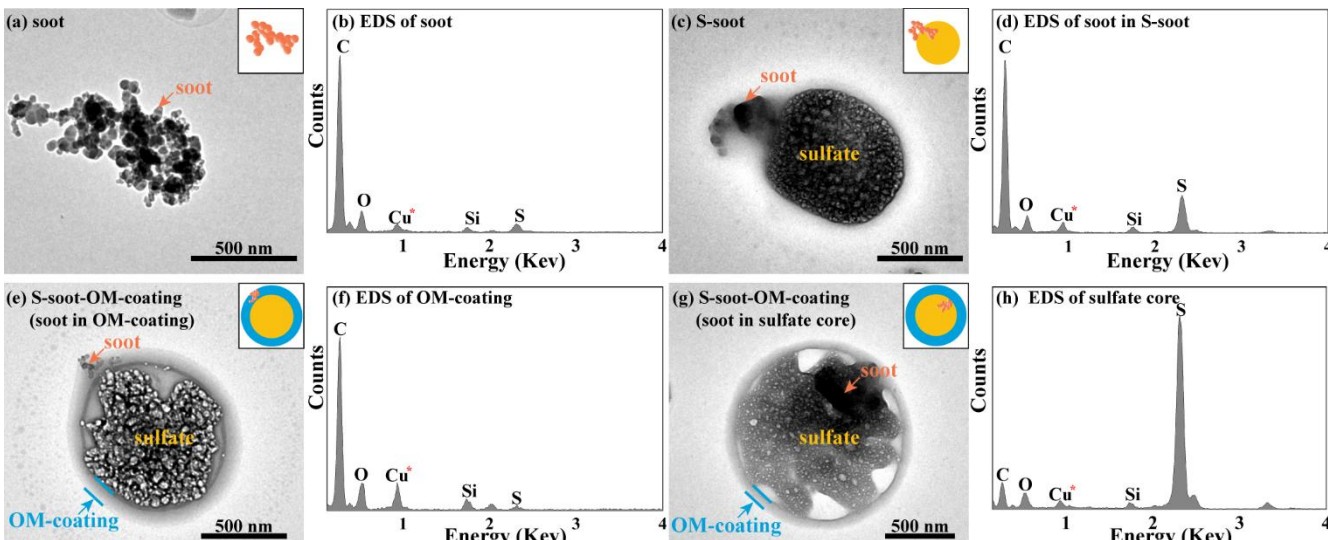

**Figure 2: The major types of soot-containing particles in this study. (a) TEM image of soot. (b) EDS spectrum of soot. Asterisk represents Cu, which was excluded from the EDS detection due to the copper material of TEM grids. (c) TEM image of S-soot. (d) EDS spectrum of soot in S-soot. (e) TEM image of the S-soot-OM-coating with soot in the OM-coating. (f) EDS spectrum of OM-coating. (g) TEM image of the S-soot-OM-coating with soot in the sulfate core. (h) EDS spectrum of the sulfate core. All conceptual graphs are placed in the top right corner of each image. The orange-circle aggregates, yellow circles, and blue rings represent soot, sulfate, and OM-coatings, respectively.**




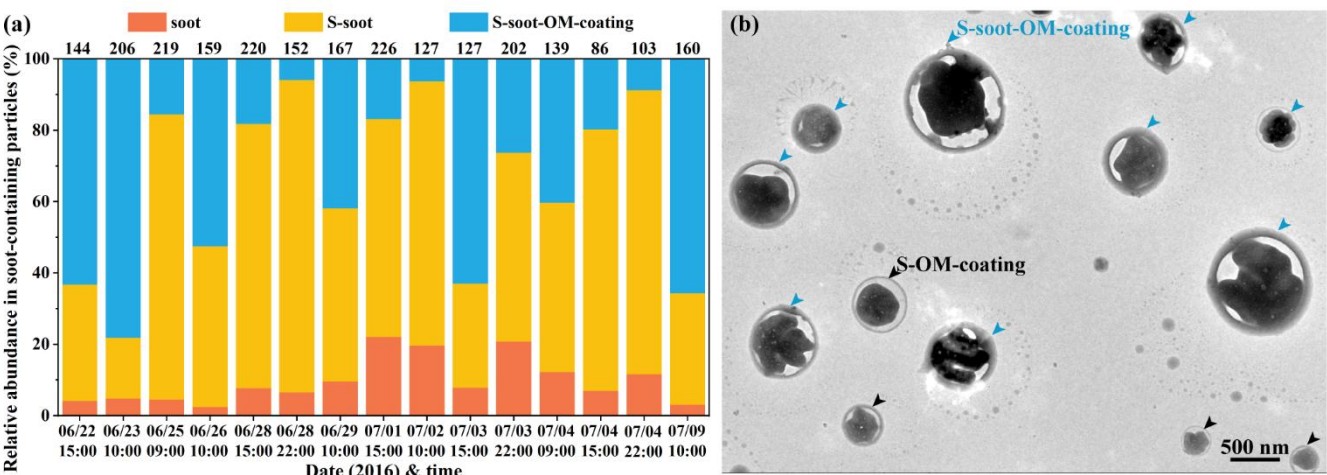

**Figure 3: (a) Relative abundance of soot, S-soot and S-soot-OM-coating particles in all soot-containing particles at the Mt. Emei site at each sampling time from 22 June to 09 July 2016. A total of 2437 soot-containing particles were analysed, and the number of analysed particles is shown above the column. (b) A typical TEM image with abundant S-soot-OM-coating particles in the sample at 10:00 on 23 June.**





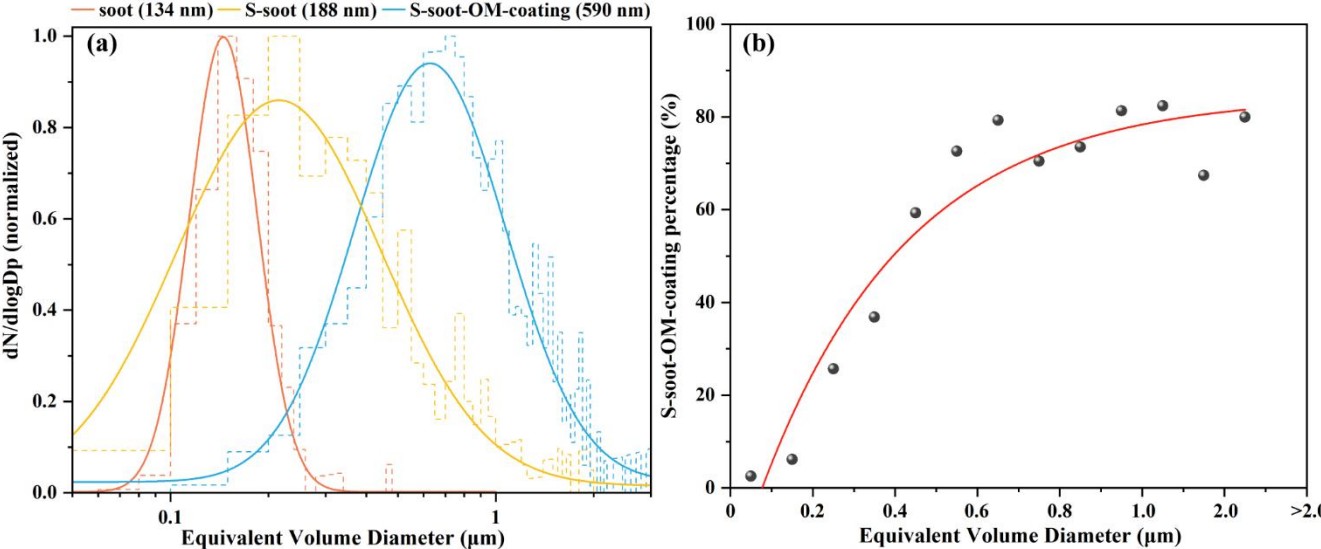


**Figure 4. (a) Size distributions of soot, S-soot and S-soot-OM-coating particles. The numbers in parentheses represent the log-normal peaks. (b) Variations in the percentage of S-soot-OM-coating particles in all soot-containing particles with sizes. The equivalent circle diameter (ECD) of individual particles was converted to the equivalent volume diameter (EVD) based on the supplemental materials.**





**Figure 5. (a)** The proportions of soot-Icore, soot-Icore-Ocoating, and soot-Ocoating particles in all S-soot-OM-coating particles. **(b)** The scatter distribution of entire particle sizes and OM-coating thicknesses of the analysed S-soot-OM-coating particles. Conceptual graphs and TEM images of **(c)** a soot-Ocoating particle, **(d)** a soot-Icore-Ocoating particle, and **(e)** a soot-Icore particle. The first and second number in each parenthesis represents the average size and OM-coating thickness of the corresponding S-soot-OM-coating particle, respectively.





**Figure 6.** OM-coating thicknesses and size distributions of soot particles with different distribution positions. (a) The different ratios of OM/soot in the S-soot-OM-coating particles. (b) Number fractions of soot-Icore, soot-Icore-Ocoating, and soot-Ocoating particles in all S-soot-OM-coating particles in different ratios of OM/soot. (c) A typical TEM image of a soot-Icore particle with two soot particles in a sulfate core (OM/soot<0.2). (d) A typical TEM image of a soot-Icore-Ocoating particle with a soot particle in a sulfate core (OM/soot≈0.1) and two soot particles in an OM-coating (OM/soot≈0.4). (e) A typical TEM image of a soot-Ocoating particle with a soot particle in the OM-coating (OM/soot≈0.5).






**Figure 7. (a)** Scatter diagram of OM/soot and the entire particle size of the S-soot-OM-coating particles. Different colours represent the number of soot particles being captured in the OM-coating. **(b)** Correlation between the average size of the S-soot-OM-coating particle and the average number of soot particles in the OM-coating. The size of the circle point represents the average ratio of OM/soot. **(c)** A typical TEM image of a S-soot-OM-coating particle with one soot particle in an OM-coating (OM/soot≈0.2, the size of S-soot-OM-coating≈336 nm). **(d)** A typical TEM image of a S-soot-OM-coating particle with three soot particles in an OM-coating (OM/soot≈0.2-0.3, the size of S-soot-OM-coating≈652 nm). **(e)** A typical TEM image of a S-soot-OM-coating particle with five soot particles in an OM-coating (OM/soot≈0.2-0.5, the size of S-soot-OM-coating≈582 nm).





**Figure 8. (a)** Fractal Dimension (*D$_f$*) of externally mixed soot, sulfate-coated soot and organic-coated soot on Mt. Emei. **(b)** *D$_f$* of externally mixed soot at different RHs. **(c)** *D$_f$* of sulfate-coated soot at different RHs. **(d)** *D$_f$* of organic-coated soot at different RHs. The parameter n in parenthesis represents the total number of soot particles analysed for each soot category to calculate *D$_f$*.





Figure 9. A conceptual model illustrating the atmospheric processes of BC on the eastern rim of the Tibetan Plateau.





**Table 1. Morphological descriptors of soot particles on Mt. Emei during noncloud periods**

| Parameters[a] | RH | soot | Sulfate-coated soot | Organic-coated soot |
|---|---|---|---|---|
| | 63%~76% | 1.82 (0.06) | 1.76 (0.06) | 1.90 (0.13) |
| $D_f$ | 93%~99% | 1.73 (0.09) | 1.99 (0.09) | 1.96 (0.08) |
| | Avg. | 1.79 (0.09) | 1.84 (0.07) | 1.95 (0.06) |
| | 63%~76% | 0.81 (0.10) | 0.87 (0.09) | 0.88 (0.08) |
| $CV$ | 93%~99% | 0.79 (0.10) | 0.87 (0.09) | 0.87 (0.08) |
| | Avg. | 0.81 (0.10) | 0.87 (0.09) | 0.87 (0.08) |
| | 63%~76% | 0.40 (0.21) | 0.39 (0.18) | 0.41 (0.20) |
| $RN$ | 93%~99% | 0.35 (0.18) | 0.42 (0.21) | 0.43 (0.21) |
| | Avg. | 0.38 (0.20) | 0.41 (0.20) | 0.42 (0.21) |
| | 63%~76% | 1.61 (0.41) | 1.62 (0.43) | 1.63 (0.43) |
| $AR$ | 93%~99% | 1.68 (0.39) | 1.60 (0.32) | 1.58 (0.40) |
| | Avg. | 1.63 (0.41) | 1.61 (0.39) | 1.61 (0.42) |

[a]$D_f$, mass fractal dimension; $CV$, convexity of soot particles; $RN$, roundness of soot particles; $AR$, aspect ratio of soot particles. In parenthesis: standard error (*s.e.*). The standard error for $D_f$ was calculated from the uncertainty in the mean-square fit considering the uncertainty in $N$ and $d_p$.