# Peer review of "Measurement report: New insights into the mixing structures of black carbon on the eastern Tibetan Plateau: soot redistribution and fractal dimension enhancement by liquid—liquid phase separation"

_Atmospheric Chemistry and Physics, 2022_

## Author Comment (AC1)

**Response to referee #1**

We are grateful for referee #1's comments. Those comments are all valuable and helpful for improving our paper and English writing. We answered the comments carefully and have made corrections in the submitted manuscript. The corrections and the responses are as following:

In the revised manuscript, the red color was marked as the revised places.

**General comments**

*This manuscript by Yuan et al. reports detailed mixing states and shapes of soot particles mixed with organic matter and sulfate collected at the eastern Tibetan Plateau. They discussed liquid-liquid phase separation and redistribution of soot particles within particles. Mixing states and shapes of soot particles influence the optical properties of internally mixed particles and their radiation. Thus the results are important to the understanding of climate influence. My major concern is that it is probable that the mixing states and shapes that they measured could be influenced by both atmospheric processes and impaction on the substrates when collected. Therefore, I suggest more careful discussion of the influence of the changes on the filter should be provided. I also recommend having some discussion based on chemical and physical processes about liquid-liquid phase separation and soot redistribution.*

**Major comments**

*Comment #1: The TEM images show mixing states after the particle collection on the substrates. Thus, changes in shapes and mixing states should be carefully discussed if they had changed in the air or on the substrate. Discussing the two-dimensional mixing states of particles on the substrate is acceptable. However, when discussing their mixing states in the atmosphere, the coating materials should cover the entire surface. When discussing the implication for the climate, the discussion should*

*depend on their three-dimensional shape in the atmosphere. At least, the TEM images show that organic coatings cover the perimeter of the spread sulfate, which cannot be a realistic 3D shape in the atmosphere. The coating thickness in the TEM image may be different from that in the atmosphere as they spread over the substrate. Sulfates are also shrinking by losing water on the substrate and have some space with organic coatings (Fig. 3b). There are also some traces around the particles (Fig. 3b). As a result, the shapes and mixing states could have been different from their original or partially the same as the These points need to be clearly discussed in the paper.*

**Reply:** We appreciate the reviewer's comments.

TEM is one of the best technique to determine real mixing state of single particles and has been widely used in large amounts of laboratory studies and field observations (Li et al., 2016). The 3D morphology of the aerosol particles in this study was investigated by using an atomic force microscope (AFM). We provide a typical AFM image of an OM-coating particle in Figure S2. As shown in Fig. S2a, there is a linear relationship between the ECD and EVD of particles and the relation between d and D is D=0.4144×d. The sizes and coating thicknesses of individual particles are calculated based on the EVD and the detailed calculation method can be found in Zhang et al. (2022).

In context, line 122-124: "As shown in Fig. S2a, there is a linear relationship between the ECD and EVD of particles with D=0.4144×d. The sizes and coating thicknesses of individual particles are calculated based on the EVD and the detailed calculation method can be found in Zhang et al. (2022)."

Line 205-206: "Similar to the method employed by Zhang et al. (2022), we calculated the OM-coating thicknesses and the entire particle sizes based on TEM and AFM (Fig. 5)."

[Figure]

55

"Figure S2. (a) The correlation of equivalent circle diameter (ECD, d) and the equivalent volume diameter (EVD, D) obtained by AFM. (b) An AFM image of an OM-coating particle."

*Comment #2: Although liquid-liquid phase separation and soot redistribution is*
60 *interesting results, I suggest having more discussion based on chemical and physical processes. Why do they have such a process? What are the physical and chemical processes (e.g., the hygroscopicity of soot, surface tension, viscosity, etc.)? When did these processes occur? Some more discussion in Fig. 9 will be helpful in interpreting the results.*

65 **Reply:** We appreciate the reviewer's comments. We conducted a solid reference survey and added the detailed explanation of liquid-liquid phase separation and soot redistribution phenomenon.

In context, line 165-171: "Optical and fluorescence microscopy analyses revealed that the LLPS could occur in individual ambient aerosols, with the presence
70 of two separate phases: inner ammonium sulfate and outer secondary organic material (You et al., 2012). Cryo-TEM measurements further confirmed that the LLPS formed the distinct core-shell structures with sulfate core and OM-coating in ambient aerosols (Altaf et al., 2016;Li et al., 2021). Furthermore, the LLPS particles have been widely observed in Arctic air (Kirpes et al., 2022;Yu et al., 2019), rural and mountain areas
75 (Zhang et al., 2022), and forest air (Li et al., 2020). Therefore, we concluded that S-soot-OM-coating particles as shown in Figure 2 can be considered as soot particles mixed with the LLPS particles."

In context, line 174-181: "It is well known that soot particles typically contains hydrocarbons, polycyclic aromatic hydrocarbons, and partially oxidized organics generated during combustion (Long et al., 2013;Wang, 2011). Moreover, TEM observations revealed a thin amorphous organic coating on carbon nanospheres of fresh soot particles (Buseck et al., 2014). The combustion processes always produce extremely thin organic layers on each soot monomer (Leskinen et al., 2023;Chen et al., 2016). Freedman (2017) showed that the LLPS process can influence surface and interfacial tensions among different phases in individual particles. Therefore, some studies used the intermolecular forces and interactions between similar chemical bonds to explain the phenomenon of soot redistribution in individual particles (Brunamonti et al., 2015;Zhang et al., 2022)."

**Specific comments**

*Comment #1: Line 52 "This uncertainty in BC radiative forcing is largely" Are you discussing an uncertainty or "a large difference in several model studies" here? Is this uncertainty caused by only "the lensing effects of the coating"? I assume that different emission inventories are also the cause of large uncertainty.*

**Reply:** We appreciate the reviewer's comments. We revised this sentence as follow:

In context, line 53-54: "This differences and uncertainties in BC radiative forcing are largely caused by the variability of the actual BC-mixing state in most models and in the ambient air (Hu et al., 2021;Zhai et al., 2022;Fierce et al., 2020;Riemer et al., 2019;Adachi and Buseck, 2013)."

*Comment #2: Line 121 "The equivalent circle diameter (ECD, d) and the equivalent volume diameter (EVD, D) were calculated according to the bearing area and bearing volume." Is soot particle included in the plot? If so, EVD cannot be related to ECD because of its fractal shape.*

**Reply:** We appreciate the reviewer's comments. Soot particles were not included in the plot. We provide a typical AFM image of an OM-coating particle in Figure S2.

[Figure]

"Figure S2. (a) The correlation of equivalent circle diameter (ECD, d) and the equivalent volume diameter (EVD, D) obtained by AFM. (b) An AFM image of an OM-coating particle."

*Comment #3: Line 151 "S4b). The sulfate core and OM-coating in secondary particles were identified as LLPS" Why? Please explain this reason.*

**Reply:** We appreciate the reviewer's comments. We added the result of EDS spectrum and provided some references to confirm the LLPS particles as follow:

In context, line 158-164: "Finally, the third type of soot-containing particle is core-shell particles with soot in either the core or the coating (Fig. 2e, 2g). The EDS spectrum shows that the coating is most likely to be organic matter (OM) with significantly higher carbon and lower sulfur content compared to the core (Fig. 2f, 2h). This similar core-shell particles have been identified as "OM-coating structure" (Li et al., 2016), which were reported in previous field observations and laboratory studies (Adachi et al., 2022;Li et al., 2021;Freedman, 2020;Li et al., 2020;Shi et al., 2008). Consequently, we called soot internally mixed within sulfate core or OM-coating as "S-soot-OM-coating" (Fig. 2e, 2g)."

In context, line 165-171: "Optical and fluorescence microscopy analyses revealed that the LLPS could occur in individual ambient aerosols, with the presence of two separate phases: inner ammonium sulfate and outer secondary organic material (You et al., 2012). Cryo-TEM measurements further confirmed that the LLPS formed the distinct core-shell structures with sulfate core and OM-coating in ambient aerosols (Altaf et al., 2016;Li et al., 2021). Furthermore, the LLPS particles have been widely

observed in Arctic air (Kirpes et al., 2022;Yu et al., 2019), rural and mountain areas (Zhang et al., 2022), and    forest air (Li et al., 2020). Therefore, we concluded that S-soot-OM-coating particles as shown in Figure 2 can be considered as soot particles mixed with the LLPS particles."

*Comment #4: Line 157 "A laboratory study and field observations have shown that LLPS can drive soot in core–shell particles from inside inorganic aerosols to outer organic aerosols, which is called the soot redistribution phenomenon" Why does it happen? Please explain this soot redistribution phenomenon in more detail.*

**Reply:** We appreciate the reviewer's comments. We provided the detailed explanation of soot redistribution phenomenon as follow:

In context, line 174-181: "It is well known that soot particles typically contains hydrocarbons, polycyclic aromatic hydrocarbons, and partially oxidized organics generated during combustion (Long et al., 2013;Wang, 2011). Moreover, TEM observations revealed a thin amorphous organic coating on carbon nanospheres of fresh soot particles (Buseck et al., 2014). The combustion processes always produce extremely thin organic layers on each soot monomer (Leskinen et al., 2023;Chen et al., 2016). Freedman (2017) showed that the LLPS process can influence surface and interfacial tensions among different phases in individual particles. Therefore, some studies used the intermolecular forces and interactions between similar chemical bonds to explain the phenomenon of soot redistribution in individual particles (Brunamonti et al., 2015;Zhang et al., 2022)."

*Comment #5: Line 182 "Therefore, we can conclude that soot redistribution in secondary particles is a common phenomenon on Mt. Emei." The results were obtained only from limited samples and periods. Therefore, it is difficult to have a general conclusion.*

**Reply:** We appreciate the reviewer's comments. We revised this sentence as follow:

In context, line 204: "Therefore, we conclude that soot redistribution in secondary particles is a common occurrence on Mt. Emei during the sampling period."

*Comment #6: Line 188 "Figure 5b shows that the entire particle size and coating thickness exhibited good correlations, suggesting that larger particles along with thicker OM-coatings can drive soot particles into the organics from the sulfate core due to LLPS." I do not think the correlation suggests the latter sentence. There is a large gap between observation and the discussion.*

**Reply:** We appreciate the reviewer's comments and we revised the content as follow:

In context, line 206-208: "Figure 5b shows that there is a certain positive correlation between the OM-coating thicknesses and the entire particle sizes, implying that larger S-soot-OM-coating particles tend to contain thicker OM-coating."

*Comment #7: Line 202 "The results suggest that the coarser particles following the thicker Omcoatings captured more soot particles in the OM-coating during the redistribution process" Why can it be concluded that it happened "during the redistribution process"? Can they simply be coagulated in the atmosphere, not "during the redistribution process"?*

**Reply:** We appreciate the reviewer's comments and we revised the content as follow:

In context, 238-239: "These results suggested that there was a higher tendency for multiple soot particles to distribute in the larger LLPS particles (Fig. 7c-e)."

*Comment #8: Line 204 "direct in situ evidence" I do not think it is direct and in situ evidence. They are obtained from the observation of filter samples.*

**Reply:** We appreciate the reviewer's comments and we revised the content as follow:

In context, line 240: "All of these observations provided evidence for soot redistribution in LLPS particles in the atmosphere over the eastern TP."

*Comment #9*: Line 205 "soot size" Is soot size provided? Fig. 4a shows that soot has a narrow size distribution. Which data should I see?

**Reply:** We appreciate the reviewer's comments. The original statement was deemed controversial and thus has been revised based on previous studies to provide clarity. The revised content is as follows:

In context, line 242: "The soot redistribution is probably governed by the entire particle size and the ratio of OM-coating thickness to soot size."

*Comment #10*: Line 210 "The average Df of externally mixed soot on Mt. Emei was 1.79±0.09 (Table 1), which was slightly higher than that on the southeastern TP (1.75±0.08) (Yuan et al., 2019), suggesting that the sources of soot particles in the eastern TP atmosphere were more complex" First, I do not understand the interpretation of "more complex." Second, values 1.79±0.09 and 1.75±0.08 essentially have no difference.

**Reply:** We appreciate the reviewer's comments. We have deleted this sentence.

*Comment #11*: Line 216 "The sulfate-coated soot and organic-coated soot particles had a higher CV (0.87 and 0.87, respectively), higher RN (0.41 and 0.42, respectively) and lower AR (1.61 and 1.61, respectively) than those of externally mixed soot (avg. CV=0.81, avg. RN=0.38, and avg. AR=1.63)." Interestingly, the sulfate-coated and organic-coated soot particles had nearly the same morphological parameters. Are they contradict the discussion of their fractal dimension in line 220? I do not see "a significant increase in fractal dimension" (line 220) when considering their error range and the plot in Fig 8a. The difference can be within an uncertainty range.

**Reply:** We appreciate the reviewer's comments. The differences of $D_f$ between sulfate-coated BC and organic-coated BC were really small. We have deleted this sentence and rewrote the content as follow:

In context, line 252-254: "The conclusion derived from all these morphological parameters was consistent with the compacted soot particles enclosed by sulfate and

organics. Indeed, several field and laboratory studies found that soot embedded with sulfate and organics could increase its compactness after coating (Wang et al., 2021;Xue et al., 2009;Saathoff et al., 2003)."

220 *Comment #12: Figure 7a. Please add a unit for the x-axis (nm).*

**Reply:** We appreciate the reviewer's comments. We modified the Figure 7.

[Figure]

Figure 7. (a) Scatter diagram of OM/soot and the entire particle size of the S-soot-OM-coating particles. Different colours represent the number of soot particles being captured in the OM-coating. (b) Correlation between the average size of the S-soot-OM-coating particle and the average number of soot particles in the OM-coating. The size of the circle point represents the average ratio of OM/soot. (c) A typical TEM image of a S-soot-OM-coating particle with one soot particle in an OM-coating (OM/soot≈0.2, the size of S-soot-OM-coating≈336 nm). (d) A typical TEM image of a S-soot-OM-coating particle with three soot particles in an OM-coating (OM/soot≈0.2-0.3, the size of S-soot-OM-coating≈652 nm). (e) A typical TEM image of a S-soot-OM-coating particle with five soot particles in an OM-coating (OM/soot≈0.2-0.5, the size of S-soot-OM-coating≈582 nm).

*Comment #13:* *Figure 9. I suggest having more discussion in Fig. 9. What do (>90%) and (>70%) mean? At high RH, I guess sulfates deliquesced and had a much larger size. I suggest adding how the liquid-liquid separation and soot redistribution occur in this figure.*

**Reply:** We appreciate the reviewer's comments. We modified Fig. 9 and added some discussion as follow:

[revised manuscript text omitted]

---

## Author Comment (AC2)

**Response to referee #2**

We are grateful for referee #2's comments. Those comments are all valuable and helpful for improving our paper and English writing. We answered the comments carefully and have made corrections in the submitted manuscript. The corrections and the responses are as following:

In the revised manuscript, the red color was marked as the revised places.

**General comments**

*The authors provide a measurement report about how black carbon (BC) is distributed within aged mixed organic/inorganic aerosol particles collected on the eastern Tibetan Plateau mountain site in July 2016. The used ground based collection on TEM grids and TEM and AFM to obtain size, mixing state and morphology. Basically, they confirm their previous result, Zhang et al. (2022), that liquid-liquid phase separation redistributes BC to the organic coatings for a wide range of relative humidities. In addition to their previous work, they deduced the fractal dimension (Df) of the BC and see a ranking with decreasing Df from externally mixed BC to sulfate coated BC to organic coated BC.*

*As the morphology of BC in internally mixed aged aerosol is clearly important for analyzing its radiative impact, I feel this measurement report should be published as it reconfirms previous work measured at different sites. However, I ask the authors to take the following comments/suggestions into account for a revised manuscript.*

**Major comments**

*Comment #1: The reader would benefit, if the connection to their previous work (Zhang et al., 2022) would be made stronger throughout the whole manuscript. For example, it remains unclear to me whether there is a significant difference in the ratio between organic coating thickness and BC size as a threshold above which the BC redistributes to the organic coating between the present study and that of Zhang et al.*

 *(2022). There, the authors came up with a ratio of 0.24, now they state this ratio is 0.2. My feeling is there is no significant difference between these thresholds (as they are somewhat arbitrary), but the authors need to discuss this.*

**Reply:** We appreciate the reviewer's comments. We have improved the data analysis of the OM/soot ratio and compared our results with that reported by Zhang et al. (2022). We also modified the Figure 6b. The detailed revision was shown as below.

In context, line 215-222: "To further explore this trend accurately, we divided the ratios into 15 bins between 0.1 and 0.4, which collectively accounted for over 80% of the total OM/soot ratios (Fig. 6b). We observed that when the OM/soot ratio was less than 0.1, all of S-soot-OM-coating particles were soot-Icore particles (Fig. 6b). As the ratio increased beyond 0.2, none of soot-Icore particles was observed, and nearly 60% of the total S-soot-OM-coating particles were identified as soot-Ocoating particles (Fig. 6b). When the ratio exceeded 0.32, more than 80% of the S-soot-OM-coating particles were identified as soot-Ocoating particles. Nearly all soot particles occurred in the OM-coating when the ratio of OM/soot was larger than 0.6 (Fig. 6b). These results suggest that soot tended to distribute into the organic coating instead of the inorganic core following an increasing ratio of OM/soot (Fig. 6b)."

In context, line 223-233: "Zhang et al. (2022) reported that the dominant type of the laboratory-generated soot-containing particles shifts from soot-Icore particles to soot-Ocoating particles when the OM/soot ratio increased from 0.04 to 0.34. Their field-observed soot-containing particles were almost soot-Ocoating particles when the OM/soot ratio exceeded 0.24. Our study at 0.32 of the OM/soot was close to their laboratory results, suggesting the reliability of our research outcomes. However, our field observation was slightly higher than the previous reported 0.24, and this discrepancy could be attributed to the considerable presence of soot-Icore-Ocoating particles in our study, which was rarely observed in Zhang et al. (2022). Over 50% of soot particles were distributed within OM-coating in all the soot-Icore-Ocoating particles (as shown in the oblique bar in Fig. 6b). Consequently, combining the soot-Ocoating (brown bar in Fig. 6b) and soot in OM-coating of soot-Icore-Ocoating

particles (oblique bar in Fig. 6b), we can infer that when the OM/soot ratio exceeds

60    0.2, most of soot (>80%) tend to distribute in organic phase in the atmosphere of Mt.

Emei during sampling period (as indicated by the organics-dominated region in Fig.

6b)."

[Figure]

"Figure 6. Variations of OM/soot ratios with different distribution positions of soot in S-soot-OM-coating particles.

65    (a) The different ratios of OM/soot in the S-soot-OM-coating particles. (b) Number fractions of soot-Icore,

soot-Icore-Ocoating, and soot-Ocoating particles in all S-soot-OM-coating particles in different ratios of OM/soot.

Oblique bar represents that soot were distributed within OM-coating in all of the soot-Icore-Ocoating particles. (c)

A typical TEM image of a soot-Icore particle with two soot particles in a sulfate core (OM/soot<0.2). (d) A typical

TEM image of a soot-Icore-Ocoating particle with a soot particle in a sulfate core (OM/soot<0.1) and two soot

70    particles in an OM-coating (OM/soot≈0.4). (e) A typical TEM image of a soot-Ocoating particle with a soot

particle in the OM-coating (OM/soot≈0.5)."

*Comment #2: My other concern is the significance of the differences they observe in*

*the fractal dimension between the different morphologies. I can see that the difference*

75    *between externally mixed BC and internally mixed BC in Df is significant. I doubt that*

*the small differences the authors see between sulfate coated BC and organic coated BC are significant. The authors need to explain in detail their uncertainty analysis for the values they provide in Table 1. While they state "The standard error for Df was calculated from the uncertainty in the mean-square fit considering the uncertainty in N and dp.", the details remain unclear to the reader. In addition, they do not comment on that Df is higher at elevated RH for sulfate coated BC compared to organic coated BC while it is the opposite at lower RH.*

**Reply:** We appreciate the reviewer's comments.

**1.** The differences of $D_f$ between sulfate-coated BC and organic-coated BC were really small. We have deleted the sentence and revised the content as follow:

In context, line 251-253: "The conclusion derived from all these morphological parameters was consistent with the compacted soot particles enclosed by sulfate and organics. Indeed, several field and laboratory studies found that soot embedded with sulfate and organics could increase its compactness after coating (Wang et al., 2021;Xue et al., 2009;Saathoff et al., 2003)."

**2.** As we used the ensemble method, the uncertainty of the $D_f$ of black carbon mainly comes from the uncertainties in the total number ($N$) and the average diameter ($d_p$) of soot monomers. $N$ can be calculated using the equation 1 as below:

$$N = k_a \left(\frac{A_a}{A_p}\right)^{\alpha} \qquad (1)$$

$A_a$ and $A_p$ can be obtained directly by analyzing TEM images. $\alpha$ and $k_a$ in this equation are determined by the overlap parameter ($\delta$). Therefore, the uncertainty of $N$ is mainly from $\delta$ of soot monomers. $\delta$ is calculated by equation 2 as below:

$$\delta = \frac{2a}{l} \qquad (2)$$

$a$ is the monomer radius and l is the monomer spacing. Note that the monomers overlap in the three-dimensional structure which can cause darkened color from gray to dark on the projection of soot particles in TEM images. We cannot figure out the lattice spacing between every pair of monomers in individual soot aggregate. We also can't obtain the diameter of every soot monomer through our manual efforts and

usually use the average diameter of several soot monomers for calculation. The quantification of this uncertainty is represented by the standard error of the slope given by the mean-square fit. Several previous studies have pointed out the uncertainty of $D_f$ (China et al., 2013;Pang et al., 2022) and used the same quantification method (China et al., 2013;Yuan et al., 2019). The uncertainties of convexity (CV), roundness (RN), and aspect ratio (AR) were calculated by standard error of all individual soot particles. We added the content as follow:

In context, line 131-135: "In this study, we employed the ensemble method to obtain a mean $D_f$ of soot particles with different mixing states (Wang et al., 2017). The uncertainty of the $D_f$ was attributed to the uncertainties in the numbers and diameters of soot monomers, which were mainly manually determined (Pang et al., 2022). The quantification of this uncertainty was expressed by the standard error of the slope given by the mean-square fit (China et al., 2013;Yuan et al., 2019)."

Line 140-141: "These morphological parameters can be calculated using the methods in China et al. (2013) and Yuan et al. (2019). The uncertainties of *CV*, *RN*, and *AR* were expressed by standard errors of these values in all individual soot particles."

**3.** Given the really small variations of $D_f$ values of organic-coated soot between different RH and the complicated mechanism of soot aging process under high RH, we deleted the discussion about the comparisons of $D_f$ values of soot between different RHs.

**Specific comments**

*Comment #1: Line 149: I suggest citing here some of the relevant lab studies, in particular also the cryo TEM work of the Freedman group as well. In particular, she showed that there is a size dependence on LLPS (e.g. Altlaf et al., 2016).*

**Reply:** We appreciate the reviewer's comments. We cited the lab study in the revised manuscript.

 In context, line 166-168: "Cryo-TEM measurements further confirmed that the LLPS formed the distinct core-shell structures with sulfate core and OM-coating in ambient aerosols (Altaf et al., 2016;Li et al., 2021)."

Line 178-179: "Freedman (2017) showed that the LLPS process can influence surface and interfacial tensions among different phases in individual particles."

 Line 193-195: "The result is similar to the previous reports that particle size plays a crucial role to influence the LLPS of individual particles (Altaf et al., 2016;Li et al., 2021)."

**Reference:**

140 Altaf, M. B., Zuend, A., and Freedman, M. A.: Role of nucleation mechanism on the size dependent morphology of organic aerosol, Chemical Communications, 52, 9220-9223, 2016.

China, S., Mazzoleni, C., Gorkowski, K., Aiken, A. C., and Dubey, M. K.: Morphology and mixing state of individual freshly emitted wildfire carbonaceous particles, Nature Communications, 4, 2122, 2013.

145 Freedman, M. A.: Phase separation in organic aerosol, Chemical Society Reviews, 46, 7694-7705, 2017.

Li, W., Liu, L., Zhang, J., Xu, L., Wang, Y., Sun, Y., and Shi, Z.: Microscopic Evidence for Phase Separation of Organic Species and Inorganic Salts in Fine Ambient Aerosol Particles, Environ Sci Technol, 55, 2234-2242, 2021.

150 Pang, Y., Wang, Y., Wang, Z., Zhang, Y., Liu, L., Kong, S., Liu, F., Shi, Z., and Li, W.: Quantifying the Fractal Dimension and Morphology of Individual Atmospheric Soot Aggregates, Journal of Geophysical Research: Atmospheres, 127, e2021JD036055, 2022.

Saathoff, H., Naumann, K. H., Schnaiter, M., Schock, W., Mohler, O., Schurath, U., Weingartner, E., Gysel, M., and Baltensperger, U.: Coating of soot and $(NH_4)_2SO_4$ particles by ozonolysis products of

155 alpha-pinene, Journal of Aerosol Science, 34, 1297-1321, 2003.

Wang, Y., Li, W., Huang, J., Liu, L., Pang, Y., He, C., Liu, F., Liu, D., Bi, L., Zhang, X., and Shi, Z.: Nonlinear Enhancement of Radiative Absorption by Black Carbon in Response to Particle Mixing Structure, Geophysical Research Letters, 48, e2021GL096437, 2021.

Wang, Y. Y., Liu, F. S., He, C. L., Bi, L., Cheng, T. H., Wang, Z. L., Zhang, H., Zhang, X. Y., Shi, Z. B.,

160 and Li, W. J.: Fractal dimensions and mixing structures of soot particles during atmospheric processing, Environmental Science & Technology Letters, 4, 487-493, 2017.

Xue, H. X., Khalizov, A. F., Wang, L., Zheng, J., and Zhang, R. Y.: Effects of coating of dicarboxylic acids on the mass-mobility relationship of soot particles, Environmental Science & Technology, 43, 2787-2792, 2009.

165 Yuan, Q., Xu, J., Wang, Y., Zhang, X., Pang, Y., Liu, L., Bi, L., Kang, S., and Li, W.: Mixing State and Fractal Dimension of Soot Particles at a Remote Site in the Southeastern Tibetan Plateau, Environmental Science & Technology, 53, 8227-8234, 2019.

Zhang, J., Wang, Y., Teng, X., Liu, L., Xu, Y., Ren, L., Shi, Z., Zhang, Y., Jiang, J., Liu, D., Hu, M.,

Shao, L., Chen, J., Martin, S. T., Zhang, X., and Li, W.: Liquid-liquid phase separation reduces
170 radiative absorption by aged black carbon aerosols, Communications Earth & Environment, 3, 128, 2022.

---

## Author Response (AR2)

**Response to referee #1**

We are grateful for referee #1's comments. Those comments are all valuable and helpful for improving our paper. We answered the comments carefully and have made corrections in the submitted manuscript. The corrections and the responses are as following:

In the revised manuscript, the red color was marked as the revised places.

General comments: The authors' replies did not answer my comments. They showed only relevant sentences in the revised manuscript. For example, I had three questions in my major comment #2, but none were answered. The same miscommunication can be applied to many of the specific comments. The most important comment, major comment #1, is misinterpreted.

**Reply: We seriously considered your comments and did more experiments to answer your question. Indeed, the revised manuscript was improved a lot.**

Major comment #1. The major comment #1 in the previous report was about three-dimensional shape (configuration) in the atmosphere. The authors' reply was about the representativeness of 3D morphology measured by TEM and did not answer my comment. My concern is that 3D shapes can be changed on the substrate from that in the atmosphere no matter how they are analyzed (e.g., AFM and TEM). Thus, a careful discussion will be necessary to interpret the 3D shape of aerosol particles in the atmosphere. For example, consider the particle configuration of the upper right image in Fig 9 (the particle with three soot particles with sulfate core and organic coating). When rotating the particle 90 degrees, one of the soot particles should be middle of the sulfate core in the 2D image. That means the particle shapes in Fig 6 c-d could be different from those in the atmosphere, but they had changed their shape on the substrate. If they had been the same organic coatings, the organic should have coated the entire surface, including over and behind the particles. Although I do not say that the particle shapes on the substrate are not useful, the shapes also have some information, including those when they were in the atmosphere. However, it cannot be directly compared to the particle shape on the substrate and those in the atmosphere.

So careful discussion should be provided here.

**Reply:** We appreciate the reviewer's comments. We undertake a carefully examination of this issue through a comprehensive and in-depth discussion.

Firstly, large amounts of studies have proved that microscopy imaging of the samples on substrate can be used to investigate the liquid-liquid phase separation (LLPS) process of the atmospheric particles. You et al. (2012) presented images of real-world samples collected on quartz fiber filters to reveal that atmospheric particles can undergo LLPS. O'Brien et al. (2015) investigated the LLPS in individual aerosol particles by collecting the samples on grid-supported carbon-filmed grids which is similar to our study. Another experiment shows that coated soot particles on the flat grid experienced minimal structural changes during the sampling and storage (Chen et al., 2017). In addition, the aerosol optical tweezer measurement provides the evidence that the core-shell particles are formed after LLPS with the variation of RH (Gorkowski et al., 2020). Therefore, the LLPS indeed occurred in aerosol particles although the aerosol particles become flat on the substrate as the below Figure.

[Figure]

**Figure 3.** Thickness maps for a mixed organic/inorganic phase (cyan) and a pure AS inorganic phase (blue) from radial scans for two different sized LSOC/AS 8:1 OIR particles at 99% RH (a, b) and two different sized HMMA/AS 1:1 OIR particles at 92% RH (c, d). The organic mixed phase is overlaid on top of the inorganic phase. Insets show STXM images for each particle taken at 288.5 eV.

Thickness maps for a mixed organic/inorganic phase (cyan) and a pure AS inorganic phase (blue) from radial scans for two different sized LSOC/AS 8:1 OIR particles at 99% RH (a, b) and two different sized HMMA/AS 1:1 OIR particles at 92% RH (c, d). This figure is cited from O'Brien et al. (2015).

Secondly, as your comments, we did additional measurements by using the same sample. Elemental maps and line scanning of C, S, O, N, and K of the typical S-soot-OM-coating particles were conducted from the TEM-EDS. The results show

the clear soot cores distributing in the OM-coating. We added the analysis method and results in the revised manuscript as follows:

In context, line 118-119: "To analyse elemental distribution in individual aerosol particles, EDS mapping and line scanning experiments were conducted using scanning TEM (STEM) mode in the JEM-2100F TEM."

Line 167-169: "Elemental mapping and line scanning analysis of the core-sell particles all reveal that the coating predominantly contains abundant C, while the inorganic core contains abundant S, N, O, and minor K (Fig. 3)."

[Figure]

"Figure 3. Elemental distribution in individual S-soot-OM-coating particles. (a) A dark-field TEM image of a S-soot-OM-coating particle. (b-f) TEM-EDS elemental maps of C, S, O, N, and K of the S-soot-OM-coating particle. (g) A TEM image of a S-soot-OM-coating particle for line scanning detection. (h) Counts per second (cps) of C and S along the pathway of line scanning over the S-soot-OM-coating particle."

Major comment #2: I am not convinced by the idea that "liquid-liquid phase separation" and "soot redistribution" processes caused the mixing states. I think that the mixing states can also be explained by coagulation between soot and sulfate (Fig. 9 left two processes) and then condensation of organic matter without LLPS and soot redistribution. I am not convinced by the process between the second left and the third left particle images (LLPS) in Fig 9. Why is the soot embedded within the sulfate core instead of simply attaching to the surface? The processes that the authors claimed may be possible, but evidence and discussion are not sufficient. In the specific comments #2, #3, #4, and #7 in my previous referee report, the authors added some more references and showed previous studies but did not discuss the results obtained in this study, including line 164-182. Although previous studies have shown LLPS processes, they do not prove that the results in the current study are the same processes. The authors did not discuss other possibilities. It is acceptable to discuss any possibility, but this manuscript has too strong a conclusion for a possibility. E.g., line 24-27 "One-third of soot-containing particles showed the LLPS phenomenon between organic matter and inorganic aerosols in individual particles, which further induced soot redistribution. The results show that a larger LLPS particle size and a higher ratio of organic coating thickness to soot size tended to drag soot from the sulfate core into the organic coating" and line 232-234 "The TEM images clearly demonstrate the transferred position of soot from the inner sulfate core to the outer organic coatings following the increasing OM/soot (Fig. 6c-e)."

Reply: We appreciate the reviewer's comments.

1. The atmospheric processes like condensation and coagulation of soot with secondary aerosols can indeed affect the morphologies of the internally mixed particles (Corbin et al., 2023). However, the coagulation process between soot and secondary particles is generally negligible except very near the source (Sedlacek et al., 2022) and is unlikely to cause the compaction of soot (Corbin et al., 2023). Condensation and other processes may participate in the aging process during long-range transport. However, here we do not concern the detail about the mixing process of soot and secondary aerosols during the long-range transport because the mixing process is very complicated and it is not the focus of this study. We focused on the structural changes of soot particles after long-range transport and influenced by

the liquid-liquid phase separation. In order to weaken the description of the related conclusions, we revised the contents as follows:

In context, line 24-27: "One-third of soot-containing particles showed a core-shell structure that probably formed the LLPS phenomenon after long-range transport. Particle size and ratio of organic coating thickness to soot size are two of the major possible factors that likely induce soot redistribution between organic matter and inorganic aerosols in individual particles."

Line 178-179: "Therefore, S-soot-OM-coating particles as shown in Figure 2 were likely considered as soot particles mixed with the LLPS particles after long-range transport."

Line 196-197: "The results indicate that a substantial amount of soot-containing particles may undergo the LLPS process during the sampling period."

Line 229-230: "These results suggest that soot exhibits a high likelihood of distributing into the organic coating instead of the inorganic core following an increasing ratio of OM/soot (Fig. 8b)."

Line 248-249: "All of these observations provided evidence supporting the potential phenomenon of soot redistribution within LLPS particles in the atmosphere over the eastern TP."

Line 277-279: "Our morphological analysis suggests that the entire particle size and ratio of organic coating thickness divided by the size of soot (OM/soot) are two of the major possible factors affecting the redistribution of soot."

Line 281-282: "Consequently, larger particles with thicker organic coatings are more inclined to drive and capture more smaller soot particles from the inorganic phase to the organic phase."

2. To avoid misleading in Fig.11. We have revised it to facilitate a better understanding of the content of this paper.

[Figure]

"**Figure 11.** A conceptual model illustrating the atmospheric processes of soot on the eastern rim of the Tibetan Plateau."

Major Comment #3: (Response to Comment #2 in the previous report): If soot particles were not included in the plot, how were the soot EVDs obtained? For example, soot EVD values are shown in Figure 4. Due to the fractal shape of soot, if soot EVD values were obtained from the plot without measuring soot particles, the soot EVD values would be inappropriate.

**Reply:** We appreciate the reviewer's comments. We did not clarify this issue in previous version. Although secondary particles and soot particles both appear to be two-dimensional in transmission electron microscopy observations (as shown in Fig. 7a, 7b), they exhibit different three-dimensional features when analyzed from other perspectives. To make it clearly, we did the surface morphology measurement by using scanning electron microscope (SEM). Sulfate and organics mostly showed a thin and flat morphology while soot was more three-dimensional (Fig. 7c, 7d). AFM analysis further show that soot exhibits a higher height than that of secondary particles (Fig. 7e, 7f). Therefore, we used ECD of soot and EVD of secondary particles to represent the individual particle size. We used the difference between the EVD of the entire core-sell particle and the EVD of inorganic core to calculate the thickness of organic coating.

In general, we modified a figure (Fig. 5), added a new figure (Fig. 7), and revised the content as follow:

In context, line 75-77: "In this study, individual particle collection, transmission electron microscopy (TEM), scanning electron microscope (SEM), and atomic force microscopy (AFM) were comprehensively employed to investigate the mixing structures of soot particles at a mountain site on the eastern fringe of the TP."

Line 121-124: "The SEM (Zeiss Ultra 55, Germany) was used to obtain detailed information on the surfaces of individual aerosol particles. To obtain three-dimensional morphological information of individual particles on the substrate, we tilted the sample stage to 75° with accelerating voltage of 10 kV and work distance of 6.6 mm, and then captured the particle images at a magnification of 7000×."

Line 130-131: "The sizes and coating thicknesses of secondary particles are calculated based on the EVD and the sizes of soot particles are calculated based on the ECD. The detailed calculation method can be found in supplement."

Line 213-214: "Similar to the method employed by Zhang et al. (2022), we calculated the OM-coating thicknesses and the entire particle sizes based on TEM and AFM (Figs. 7, S2, S3)."

[Figure]

"Figure 5. (a) Size distributions of soot, S-soot and S-soot-OM-coating particles. The numbers in parentheses represent the log-normal peaks. (b) Variations in the percentage of S-soot-OM-coating particles in all soot-containing particles with sizes."

[Figure]

"Figure 7. Different views from the different microscopic techniques. (a) A TEM image of S-soot-OM-coating particle, (b) A TEM image of soot particles, (c) A SEM image at 75° tilt angle of individual particles, (d) A SEM image of soot particles, (e) An AFM image of S-soot-OM-coating particle, (f) The cross-sectional analysis from the left AFM image."

Specific Comment #1: OM/Soot

Figure 6. How were the OM/soot values obtained for each soot particle within particles in Fig. 6 c-e?

**Reply:** We appreciate the reviewer's comments. Firstly, we obtained the ECD of the entire S-soot-OM-coating particle (ECD1, blue line, Fig. S3) and ECD of the inorganic core (ECD2, green line, Fig. S3) through the Radius software. Secondly, we transform the ECD1 and ECD2 to EVD1 and EVD2 according to the AFM analysis, respectively. Thirdly, we calculate the OM-coating thickness (OM for short,

(EVD1-EVD2)/2). Finally, we calculate the OM/soot ratio (OM/ECD3).

[Figure]

"**Figure S3**. The calculation diagram of the ratio of OM coating thickness to soot size (OM/soot) of a typical S-soot-OM-coating particle. OM/soot = [0.4144(ECD1-ECD2)/2]/ECD3"

Lines 268-270: "Once the OM/soot ratio exceeded 0.2, more than 80% of the soot tended to be distributed in the organic coating due to the possible intermolecular forces and interactions with increasing coating thickness (Figs. 6b, 9)". The OM/soot ratio >0.2 means that there is more OM compared to the soot, indicating that the soot tends to distribute in the organic coating regardless of "the possible intermolecular forces and interactions with increasing coating thickness".

**Reply:** We appreciate the reviewer's comments. We revised this sentence as follow:

In context, line 279-280: "Once the OM/soot ratio exceeded 0.2, more than 80% of the soot tended to distribute in the organic coating (Figs. 8b, 11)."

**References**

Chen, C., Enekwizu, O. Y., Ma, Y., Zakharov, D., and Khalizov, A. F.: The Impact of Sampling Medium and Environment on Particle Morphology, Atmosphere, 8, 162, 2017.

Corbin, J. C., Modini, R. L., and Gysel-Beer, M.: Mechanisms of soot-aggregate restructuring and compaction, Aerosol Science and Technology, 57, 89-111, 2023.

Gorkowski, K., Donahue, N. M., and Sullivan, R. C.: Aerosol Optical Tweezers Constrain the Morphology Evolution of Liquid-Liquid Phase-Separated Atmospheric Particles, Chem, 6, 204-220, 2020.

O'Brien, R. E., Wang, B. B., Kelly, S. T., Lundt, N., You, Y., Bertram, A. K., Leone, S. R., Laskin, A., and Gilles, M. K.: Liquid-Liquid Phase Separation in Aerosol Particles: Imaging at the Nanometer Scale, Environmental Science & Technology, 49, 4995-5002, 2015.

Sedlacek, A. J., III, Lewis, E. R., Onasch, T. B., Zuidema, P., Redemann, J., Jaffe, D., and Kleinman, L. I.: Using the Black Carbon Particle Mixing State to Characterize the Lifecycle of Biomass Burning Aerosols, Environmental Science & Technology, 56, 14315-14325, 2022.

You, Y., Renbaum-Wolff, L., Carreras-Sospedra, M., Hanna, S. J., Hiranuma, N., Kamal, S., Smith, M. L., Zhang, X., Weber, R. J., Shilling, J. E., Dabdub, D., Martin, S. T., and Bertram, A. K.: Images reveal that atmospheric particles can undergo liquid–liquid phase separations, Proceedings of the National Academy of Sciences, 109, 13188-13193, 2012.

---

## Author Response (AR3)

**Response to referee #1**

We are grateful for referee #1's comments. Those comments are all valuable and helpful for improving our paper. We answered the comments carefully and have made corrections in the submitted manuscript. The corrections and the responses are as following:

In the revised manuscript, the red color was marked as the revised places.

Comment #1: Line 75: Change "scanning electron microscope" to "scanning electron microscopy"

**Reply:** We appreciate the reviewer's comments. We revised the word in the manuscript.

In context, line 75: "In this study, individual particle collection, transmission electron microscopy (TEM), scanning electron microscopy (SEM), and atomic force microscopy (AFM) were comprehensively employed to investigate the mixing structures of soot particles at a mountain site on the eastern fringe of the TP."

Comment #2: In Fig. 7, the SEM images (Fig 7c and d) look different. At least the type of substrate is different. Are they the same samples?

**Reply:** We appreciate the reviewer's comments. The SEM images presented in Fig. 7c and 7d were obtained from different samples collected during the sampling period on Mt. Emei. Our intention was to visually demonstrate the distinct surface morphologies of sulfate, organic, and soot particles through SEM analysis. We added the detailed information in the method section.

In context, line 105-108: "A DKL-2 sampler (Genstar Electronic Technology, China) was used to collect individual aerosol particles on copper TEM grids covered by carbon film (carbon type-B, 300-mesh copper; Tianld Co., China) and silicon wafers (thickness: $500\pm10$ μm, size: $3\times3$ mm; LIJINGKEJI, China). We also collected individual aerosol particles onto 47 mm diameter polycarbonate filter membranes (600 nm pore size, Whatman Inc., USA) via a Mini-Vol Sampler (Airmetric, USA) for SEM analysis."

In context, line 111-112: "The copper grids, silicon membranes, and

polycarbonate filter membranes were stored in the dry, clean, and airtight containers at 20-25% RH until analysis."